

# Impact of anthropogenic activities on global land oxygen flux

Xiaoyue Liu[1], Jianping Huang[1]*, Jiping Huang[2], Changyu Li[1], Lei Ding[1]
[1]Key Laboratory for Semi-Arid Climate Change of the Ministry of Education, Lanzhou University, Lanzhou, 730000, China
[2]Enlightening Bioscience Research Center, Mississauga, L4X 2X7, Canada

*Correspondence to*: Jianping Huang (hjp@lzu.edu.cn)

**Abstract.** Atmospheric oxygen ($O_2$) is one of the predominant features that enable earth as a habitable planet for active and diverse biology. However, observations since the late 1980s indicate that $O_2$ content in the atmosphere is falling steadily at part-per-million level. Although a scientific consensus has emerged that the current decline is generally attributed to the combustion of fossil fuel, a quantitative assessment of the anthropogenic impact on the $O_2$ cycle on both global and regional
scale is currently lacking. This paper quantifies the anthropogenic and biological $O_2$ flux over land and provides a quantitative and dynamic description of land $O_2$ budget under impacts of human activities on a grid scale. It is found that total anthropogenic $O_2$ flux over land has risen from 35.6 Gt/yr in 2000 to 46.0 Gt/yr in 2013, while the production from land (11.5Gt/yr averaged from 2000 to 2013) displays a faint increase during the same period. High anthropogenic fluxes mainly occur in Eastern Asia, India, North America and Europe caused by fossil fuel combustion and in Central Africa caused by
wildfire. Due to strong heterotrophic soil respiration under higher temperature conditions, the positive biological $O_2$ flux in the tropics is not significant. Instead, boreal forest and Tibetan plateau become the most important sources of atmospheric $O_2$ in the Anthropocene. The anthropogenic oxygen consumption data are publicly available online at https://doi.org/10.1594/PANGAEA.899167.

## 1 Introduction

In recent decades, the strong response that global carbon cycle gives to anthropogenic forcing is addressed by voluminous literature on both global and regional scale (Huang et al., 2007; IPCC, 2013; Luyssaert et al., 2007), which is essential for understanding environmental history of our planet and predicting and guiding our future. However, the $O_2$ cycle, another fundamental biogeochemical cycle in the earth system, is also responding forcefully to global change with its own dynamic. The decline of $O_2$ concentration, an average loss rate of 4 ppm per year, is comparable in magnitude to those of $CO_2$. The
essentiality of atmospheric $O_2$ to the habitability of our planet Earth and the survival of humankind can never be overstated since an equable $O_2$ in the atmosphere is central to life (Petsch, 2013). Therefore, systematic investigations on $O_2$ cycle, especially under anthropogenic forcing deserves to be high on the agenda. Fossil fuel combustion which effectively causes an $O_2$ loss has clearly not been identified as the only culprit responsible for the recent decline. Additional processes including respiration of human and livestock, wildfire, production from terrestrial and oceanic ecosystems also play a part
(Huang et al., 2018).



Most of the existing concepts of $O_2$ budget are based on a static scenario, which assumes that atmospheric $O_2$ concentration remains constant and each component of the budget is independent of time (Bender et al., 1994a, 1994b). Nevertheless, since the beginning of the Industrial Era, humans have been producing energy by combusting fossil fuels, a process that not only emits considerable amounts of $CO_2$ into the atmosphere (IPCC, 2013), but also removes substantial amount of molecular $O_2$

from the atmosphere, thus resulting in the disturbance of both carbon and $O_2$ cycle comparable to the major natural flux in magnitude. Records from polar ice core indicate a decline of atmospheric $O_2$ related to the burning of fossil fuel (Battle et al., 1996). Previous research has shown that approximately half of the anthropogenic $CO_2$ since the Industrial Era has been absorbed by the ocean and terrestrial ecosystem (Quéré et al., 2018), while the response of each component involving the $O_2$ cycle to anthropogenic forcing is poorly discussed. Does more $O_2$ transport from land and ocean to the atmosphere to

compensate for the current depletion of $O_2$? What is the contribution of land ecosystem to the supplement of atmospheric $O_2$? How will the contribution variate in the future? These questions remain to be answered systematically.

Additionally, the descriptions of $O_2$ budgets are generally provided in a global mean, only roughly representing the globally averaged value of each process. However, the amount of $O_2$ consumed and produced is not uniform over land and could vary dramatically from place to place due to different levels of human activity (burned fuel type, population density etc.) and

different natural conditions (vegetation cover, growing season etc.). Therefore, in some regions, the $O_2$ consumption may far exceed the production, which requires the supplementary $O_2$ from regions where production of $O_2$ surpasses the consumption to maintain the local $O_2$ level through atmospheric circulation. Where does the supplementary $O_2$ come from and how much $O_2$ can it provide? Will the $O_2$ supply from these areas increase or decrease in the future? To address those interesting problems, a dynamic global $O_2$ budget on a grid scale is necessary.

In this paper, most major types of anthropogenic $O_2$ consumption processes are estimated globally on each grid and are combined with the estimation of terrestrial biological $O_2$ flux to provide a dynamic $O_2$ budget on a grid scale that varies with space and time. The dynamic $O_2$ budget can provide us with an insightful perspective on global climate change.

## 2 Datasets and Methods

### 2.1 The observational oxygen concentration data

In this study, $O_2$ concentrations of 9 stations from the Scripps $O_2$ Program ([http://scrippso2.ucsd.edu/](http://scrippso2.ucsd.edu/)) are used. Since these data from the 9 stations are collected from distant locations, they are able to represent the average concentrations over a wide range of areas rather than local background values (Keeling, 2018).

The concentration of atmospheric $O_2$ is expressed as changes in the $O_2/N_2$ ratio of air relative to a reference (air collected in the mid-1980s). The observed changes are so tiny that they are reported in per meg units (Manning and Keeling, 2006):

$$\delta = ((O_2/N_2)_{sample} - (O_2/N_2)_{reference})/(O_2/N_2)_{reference} \times 10^6 \tag{1}$$

where the subscripts 'sample' and 'reference' denote the sample air and the reference air collected in the mid-1980s, respectively. 1 per meg equals $10^{-6}$. The conversion between volume fraction (ppm and per meg) and mass (Gt $O_2$) can be



achieved by the following equation: 1 per meg = 0.20946 ppm = $M \times 10^{-6} \times 32$ g/mol $O_2$ = 1.186 Gt $O_2$, where $M = 3.706 \times 10^{19}$ moles is a reference value for the total number of $O_2$ molecules in the atmosphere.

## 2.2 The estimation of anthropogenic oxygen flux

In this paper, the following four processes including fossil fuel combustion, human respiration, livestock respiration and
wildfire are discussed. Other processes are tiny enough to be neglected, compared with the processes mentioned above, like the oxidative weathering process (Haynes et al., 2016). In terms of the wildfire consumption, some fire activities are caused naturally, while others such as agriculture waste burning, deforestation are human-induced. For convenience, we include wildfires in the calculation of anthropogenic fluxes and exclude it from the biological $O_2$ fluxes by terrestrial vegetations.

### 2.2.1 Oxygen flux caused by consumption by fossil fuel combustion

The estimation of $O_2$ flux by fossil fuel combustion is converted from $CO_2$ emissions data from the Carbon Dioxide Information Analysis Center (CDIAC, http://cdiac.ess-dive.lbl.gov/). These data provide time series of annual $CO_2$ emissions from anthropogenic sources, including fossil fuel burning, cement manufacturing and gas flaring in oil fields as well as energy production, consumption and trade data on a 1°×1° grid from 1751 to 2013.

The $O_2$ consumption in the combustion process is estimated according to the equation as follow:

$$C_xH_y + \left(x + \frac{1}{4}y\right)O_2 \rightarrow x\,CO_2 + \frac{1}{2}yH_2O.$$

Due to the different fuel mix in each country, the oxidative ratio (the number of $O_2$ moles that are consumed per mole $CO_2$ emitted) can vary over spatial and temporal scales (Steinbach et al., 2011). In this paper, the carbon emission by different types of fossil fuel burned in each country is also obtained from the CDIAC. Considering there are several historical events (collapse of the Soviet Union in 1991, independence of South Sudan in 2010, etc.) that change the national boundaries, we
adopt CShapes (Weidmann et al., 2010), a new dataset that provides historical maps of country boundaries in the post-World War II period to map the gridded oxidative ratio on country level. Then, $O_2$ consumption by fossil fuel on each is estimated based on the following equation:

$$C_{FF} = M_{O_2} \times \sum_{i=1}^{4} \frac{E_{FF_i}}{M_C} \times ratio_i \tag{2}$$

where $C_{FF}$ is the annual $O_2$ consumption (Gt $O_2$/yr); $M_{O_2}$ is the relative molecular mass of $O_2$ (32g/mol); subscript $i$ is used
to indicate the type of fuel; $E_{FF_i}$ is the $i$-th species carbon emissions from fossil fuel combustion (Gt C/yr); $M_C$ is the relative molecular mass of carbon (12 g/mol); $ratio_i$ is the molar ratio of $O_2$:$CO_2$ when the $i$-th fuel is completely burned (Table 1).



### 2.2.2 Oxygen flux caused by human respiration

The $O_2$ flux by human respiration is based on the population density datasets at $0.5° \times 0.5°$ resolution from 1980 to 2100 from Murakami and Yamagata (2016). The data can be obtained in http://www.cger.nies.go.jp/gcp/population-and-gdp.html. The datasets also provided actual GDPs from 1980 to 2010, and estimated GDPs under different SSPs from 2010 to 2100.

We assume that in a day a man works 8 hours with oxygen consumption rate at 1.0 L $O_2$/min and rests for the remaining 16 hours with oxygen consumption rate at 21.0 L $O_2$/h. Then, an adult consumes approximately 1.17 kg (816 L) of $O_2$ per day. It is estimated that an astronaut consumes 0.84kg $O_2$/day for survival (Jones, 2003). Our estimation (1.17kg) is relatively higher, which is reasonable because the astronauts have fewer physical activities than people on earth. We estimate the total respiration consumption based on the following equation:

$$C_{RES-H} = P \times C_d \times 365 \tag{3}$$

where $C_{RES-H}$ is the annual $O_2$ consumption of human respiration (unit: Gt $O_2$/yr), $P$ is the total population, and $C_d$ is the daily $O_2$ consumption per capita (kg $O_2$/day).

### 2.2.3 Oxygen flux caused by livestock respiration

$O_2$ consumption by livestock respiration is based on the global datasets on the geographic distribution of livestock from Gridded Livestock of the World v3.0 (GLW 3) (Gilbert et al., 2018). GLW 3 provides global population densities of cattle, buffaloes, horses, sheep, goats, pigs, chickens and ducks at a spatial resolution of 0.083 degrees (approximately 10km at the equator). The dataset contains 2 versions of density pattern. In the first version of dasymetric weighting, livestock numbers are distributed within census polygons according to weights established by statistical models using high-resolution spatial covariates. In the second version of areal weighting, animal numbers are distributed homogeneously with equal densities within their census polygons. Here, the second version of data is used.

The basal metabolic rate (BMR) is the rate of energy expenditure per unit time by endothermic animals at rest. It can be reported in mL $O_2$/min. The BMR (mL $O_2$/h) of a mammal can be predicted by the formula given by Kleiber (1932). BMR = $3.43M^{0.75}$, where $M$ is the animal's mass (g). Then, following the formula below, the annual $O_2$ consumption of the livestock can be estimated.

$$C_{RES-L} = \sum_{i=1}^{6} P_i \times BMR_{d_i} \times 365 \tag{4}$$

where $C_{RES-L}$ is the annual $O_2$ consumption of livestock (unit: Gt $O_2$/yr), $P_i$ is the total number of $i$-type livestock, and $BMR_{d_i}$ is the average daily $O_2$ consumption of the $i$-type livestock (unit: kg $O_2$/day).

Since the data only describes the situation in 2010, we assume that the total number of all livestock is proportional to the total human population and use the spatial pattern of the data when estimating the $O_2$ consumption in other years.



### 2.2.4 Oxygen flux caused by wildfires

The data on $O_2$ consumption by fire are estimated based on carbon emissions data derived from the Global Fire Emissions Database (GFED) (Van Der Werf et al., 2017). Satellite information on fire activity and vegetation productivities around the world are combined together to estimate gridded burned area and pollutant emissions. The version of the dataset used in this paper is version 4, with a spatial resolution of 0.25 degrees from 1997 to 2016. The GFED data classifies fire types into the following categories: savanna, boreal forest, temperate forest, tropical forest, peat and agricultural waste. Since the combustion products are all organic, the molar ratio of $O_2$: $CO_2$ is 1.1:1. The $O_2$ consumption can be estimated by the following formula:

$$C_{FIRE} = M_{O_2} \times \sum_{i=1}^{6} \frac{DM_i \times CC_i \times EF_i}{M_{CO_2}} \times 1.1 \tag{5}$$

where $C_{FIRE}$ is the $O_2$ consumption (Gt $O_2$/yr); subscript $i$ is used to indicate the type of fire; $DM_i$ is the mass of the $i$-th fire type dry matter emitted (kg DM/yr); $CCi$ is the percentage of carbon in the $i$-type fire dry matter; $EF_i$ is the emission factor (kg/g) of the $i$-th fire type $CO_2$; 1.1 is the molar ratio of $O_2$:$CO_2$ at the time of complete combustion.

### 3 Results

#### 3.1 Estimation of $O_2$ consumption by fossil fuel combustion

It's widely recognized that $O_2$ consumption by fossil fuel is the most important attribution to the recent $O_2$ decline (Keeling and Manning, 2014; Martin et al., 2017; Valentino et al., 2008). Figure 1 shows the global pattern of $O_2$ consumption by fossil fuel combustion and oxidative ratio in 2013. It is found that the patterns in $O_2$ consumption (Fig. 1a) generally follow the pattern of $CO_2$ emission. High $O_2$ consumption areas (greater than 5.0 kg $O_2/m^2$/yr) are in Eastern Asia, India, Europe and the US, while low $O_2$ consumption can be seen in South America, Africa and Australia. In the oxidative ratio map (Fig. 1b), low oxidative ratio, signifying coal as the major fuel source, are located in Eastern Asia, India and South Africa. High oxidative ratio (greater than 1.45) areas are found in Russia, Central Asia, Canada and Argentina, indicating gas as the main source of fossil fuel burning. China, United States, India and Russia are the 4 countries that consume the most $O_2$ by fossil fuel combustion. Solid fuel, which accounts for about 70% of the total carbon consumption, is the biggest source of anthropogenic $CO_2$ emission as well as the biggest anthropogenic sink of atmospheric $O_2$ in China and India. In Russia, the most important energy source is gas fuel, followed by solid fuel and liquid fuel, which leads to the high oxidative ratio (1.61). Figure 2 shows the long-term trends of global total $O_2$ consumption by fossil fuel combustion and global averaged oxidative ratio from 1975 to 2013. Global total $O_2$ consumption has increased from 17.0 Gt to 35.0 Gt during 1975~2013. The growth could be explained by the rise of solid fuel to a large extent. The oxidative ratio has experienced a significant decrease after the 21st century and reached 1.34 in 2013, which can be explained by a robust increase in the coal burning as well as the cement production in countries such as China and India. This oxidative ratio calculated in this paper is close to the



estimation from Keeling (1988) (red line), Manning and Keeling (2006) (grey shade) and COFFEE dataset provided by Steinbach et al. (2011) (from 1.39 to 1.42).

Long-term trends of $O_2$ consumption by fossil fuel combustion and oxidative ratio from 1975 to 2013 are shown in figure 3. Figure 3a shows the global trend of $O_2$ consumption on a gridded level, in which the warm colors indicate an increase and cold colors a decrease in $O_2$ consumption on each grid. The increase mainly occurs in East Asia and India, while the decrease is located in Europe. Figure 3b shows the trend of the oxidative ratio. With the increasing demand for solid fuels, the oxidative ratio displays a downward trend globally, except for regions such as Russia, Europe and Argentina.

### 3.2 Estimation of $O_2$ consumption by human and livestock respiration

When estimating the $O_2$ consumed by human respiration, we assume that a man works about 8 hours a day and rest for the remaining 16 hours, and ignore the human activities which consume more $O_2$ such as sports and hard physical labor. As for the livestock, eight types of main livestock, including goats, ducks, buffaloes, sheep, horses, cattle, pigs and chickens are considered. The estimations are based on the basal metabolic rate, which neglects the metabolic enhancement caused by ambient temperature variation, exercise, etc (Cai et al., 2018). Therefore, the actual $O_2$ consumption by human and livestock would be greater than the estimation in this study.

Figure 4 shows the global pattern and long-term trends of $O_2$ consumed via human respiration. The highest $O_2$ consumption occurs in India and eastern China, up to 0.7 kg $O_2/m^2/yr$, which are the most populated regions around the world (Figure 4a). The population has grown worldwide, especially in India and East Asia. The global distribution of $O_2$ consumption (Figure 5a) follows a similar pattern because the livestock industry should be well-developed to cope with food demand from high population density. Among the 5 continents (Asia and Australia are considered as one continent in this paper because Australia is relevantly small in both area and $O_2$ consumption) illustrated in Figure 5, buffaloes, cattle, chickens, ducks, goats, pigs and sheep in Asia and Australia consumes the most $O_2$, while horses in South America account for the most $O_2$ consumed by livestock.

In terms of the global total volume, the $O_2$ consumed by livestock and human respiration is roughly equivalent (Figure 6), about 3.0 Gt in 2013 and the total $O_2$ consumed by human and livestock has increased from 3.4 Gt to 6.0 Gt during 1975-2013. Among the 8 types of the main livestock around the world, cattle consume the most $O_2$, accounting for 55% of total $O_2$ consumed by livestock.

### 3.3 Estimation of $O_2$ consumption by wildfire

Figure 7 shows the global distribution of $O_2$ consumption by wildfire from 1997 to 2016. The highest consumption by wildfire is mainly located in the tropical region, especially central Africa since these areas are rich in surface vegetation and the net primary productivity (NPP) is relevantly high. The burning of these areas would emit more carbon and thus consume more $O_2$.





Global mean $O_2$ consumption is 5.87 Gt $O_2$/yr and shows a weak downtrend during 1997–2016, with a maximum of 8.1 Gt $O_2$/yr in1997 and minimum of 4.8 Gt $O_2$/yr in 2013. In terms of $O_2$ consumption of various types of fires, savanna fires have the widest distribution and consume the most $O_2$, accounting for 60%~70% of the total; followed by fires in tropical forests.

**3.4 Global land net $O_2$ flux**

The global distribution of land $O_2$ flux from 2000 to 2013 is illustrated in figure 8a. In figure 8a, total anthropogenic flux, including fossil fuel combustion, respiration of human and livestock, and wildfire are drawn. The distribution of total $O_2$ consumption generally follows the pattern of $O_2$ consumption by fossil fuel combustion in most part of the world, since, among the four $O_2$ consumption processes, combustion of fossil fuels consumes the most $O_2$ and are mainly distributed in areas with a high density of livestock and population. There are several significantly high $O_2$ consumption area around the

world: East Asia, India, eastern North America, Europe and central Africa. Except for central Africa, these areas are densely populated with high levels of human activities and thus consume more $O_2$ than other areas of the world, while in central Africa, where the wildfire accounts for the vast majority of local $O_2$ consumption, also exhibits a high level of anthropogenic $O_2$ flux.

The net biological $O_2$ flux over land are estimated based on NEE (Net Ecosystem Exchange) simulated by the CASA model,

with fire activities excluded. Fire activities are considered as the anthropogenic fluxes in this paper. Negative flux (brown regions) indicate places where uptake of $O_2$ from the atmosphere occurs. Positive flux (green colors) indicate places where the production of $O_2$ occurs. Globally, most regions show positive flux, which means the terrestrial ecosystem is transporting $O_2$ to the atmosphere in most parts of the world. It is worth noting that in some tropical regions such as Southeast Asia and Amazon forest, the biological $O_2$ flux is negative, but the total $O_2$ flux over tropical land areas (including South American

Tropical, Tropical Asia and North Africa) shows positive flux of 2.1Gt/yr. In Northern Temperate (Temperate America and Temperate Eurasia), total biosphere flux amounts to 3.0Gt/yr (Figure 10).

The spatial distribution of the difference between anthropogenic $O_2$ flux and biological $O_2$ flux, which is defined as the land net $O_2$ fluxes, averaged from 2000 to 2013 is shown in figure 8c. The brown regions, mainly located in eastern Asia, Europe, North America, and northern South America, covering more than half of the land, denote the areas where anthropogenic $O_2$

flux exceeds the biological $O_2$ flux. In these areas, humans are consuming more $O_2$ than the ecosystem can provide. The area shaded with green color, including northern Canada and Siberia, represents the region that can still emit $O_2$ into the atmosphere under the impact of human activities. However, it should be noted that since the positive $O_2$ flux is comparably smaller, the color bar, which exhibits larger negative values and smaller positive values, has been modified to enhance visualization, so that readers can observe the positive $O_2$ flux from the land more closely.

Due to the increasing fossil fuel combustion, overgrazing and population growth, current $O_2$ consumption over land is far greater than $O_2$ production from the terrestrial ecosystem, breaking the atmospheric $O_2$ balance and causing the decline of $O_2$ concentration in the atmosphere. During the period of 2000~2013, it is estimated that total anthropogenic $O_2$ flux over land



has risen from 35.6 Gt/yr in 2000 to 46.0 Gt/yr in 2013, while the production from land (11.5Gt/yr averaged from 2000 to 2013) displays a faint increase during the same period (Figure 9).

As for the air-sea $O_2$ flux, the possibility that the ocean might be a long-term source of $O_2$ in the atmosphere has already been widely recognized. Human-induced global warming and climate change reduce the solubility of the ocean (Bopp et al., 2002; Plattner et al., 2002) and thus causes the decline in the dissolved $O_2$ in the upper ocean and $O_2$ is released from the ocean to the atmosphere. The process above is believed to be superimposed on a natural background air-sea $O_2$ fluxes of different time scales. It is estimated that about 1.4Gt $O_2$ is outgassed from ocean per year during 2000~2010 (Keeling and Manning, 2014), which is very small compared to the magnitude of the other processes we estimated earlier, and it is difficult to provide a spatial distribution because of sparse coverage of measurements on the ocean (Keeling et al., 2010).

Since the intensity of human activities and vegetation coverage can vary over spatial and temporal scales, the local $O_2$ budget in different parts of the world can be quite different. Figure 10 shows the total land $O_2$ flux in tropical, northern temperate, northern boreal and southern temperate regions. Except for the northern boreal region, the anthropogenic $O_2$ consumption in other areas is greater than the biological $O_2$ flux that transport $O_2$ to the atmosphere. In the tropics, the biological $O_2$ flux is not high, which may be caused by strong heterotrophic soil respiration in relevantly higher temperature. In terms of the anthropogenic flux, wildfires induced $O_2$ flux is comparable to consumption of fossil fuel combustion, but they follow opposite trends over time: the fossil fuel $O_2$ flux experienced a steadily growing trend while the wildfire flux indicates a downward trend, which may be related to the reduction in wildfire activities in recent decades (Arora and Melton, 2018). In the temperate regions of the northern hemisphere, which is the most populated region in the world, total anthropogenic $O_2$ flux account for more than half of the global anthropogenic $O_2$ flux, reaching 23.5Gt/yr, and its growth rate is also the most significant among the 5 regions. In the boreal region of the northern hemisphere, wildfire activities are frequent and consume more $O_2$ than fossil fuel combustion. Since this region is sparsely populated, the land terrestrial ecosystem is playing a leading role in the local $O_2$ budget and thus become the source of $O_2$ as well as a sink of $CO_2$ in the background of global change. In the temperate of the southern hemisphere, livestock respiration consumes more $O_2$ than human respiration and the consumption of $O_2$ by wildfire also surpasses the consumption by fossil fuel due to the relevantly weak intensity of human activities. As the earth's third pole, the Tibetan plateau is one of the most important areas of global weather and climate change (Ma et al., 2017). Figure 10e shows the $O_2$ budget over the Tibetan Plateau. The biological fluxes that produce $O_2$ to the atmosphere display no significant trend while the $O_2$ consumed by fossil fuel combustion continued to rise during the years. The Tibetan plateau is another source of atmospheric $O_2$ under the impact of human activities.

**4 Data availability**

The anthropogenic oxygen consumption dataset from 1975 to 2013 is available on PANGAEA at https://doi.org/10.1594/PANGAEA.899167 as NetCDF files (.nc) with 1.0°×1.0° spatial resolution(Liu et al., 2019).



## 5 Conclusions and Discussions

We have presented a global gridded dataset of $O_2$ consumption by anthropogenic processes, including fossil fuel burning, wildfire and respiration of human and livestock based on fossil fuel carbon emission from CDIAC, carbon emission from GFED and population and livestock density data. Combining this data with biological $O_2$ flux over land, the global $O_2$ budget on land is estimated from 2000 to 2013 on a 1°×1° grid.

The $O_2$ consumption is converted from $CO_2$ emission via the estimation of the oxidative ratio of different countries. Low oxidative ratios, signifying coal as the major fuel source, are located in eastern Asia, India and Southern Africa, while a high oxidative ratio (greater than 1.45) areas are found in Russia, Central Asia Canada and Argentina, indicating gas fuel as the main source of fossil fuel burning. The oxidative ratio shows a downward trend from 2000, possibly due to the growing contribution from coal burning with low oxidative ratio and cement production that does not consume $O_2$. High consumption areas are mainly located in eastern Asia, India, North America and Europe. In terms of total volume, global total $O_2$ consumption by fossil fuel has increased from 17.0 Gt to 35.0 Gt during 1975~2013.

The $O_2$ consumed by livestock and human respiration is roughly equivalent, about 3.0 Gt in 2013 and the total $O_2$ consumed by human and livestock has increased from 3.4 Gt to 6.0 Gt during 1975-2013. Cattle consume the most $O_2$ among the 8 types of the main livestock calculated, accounting for 55% of total $O_2$ consumed by livestock.

As for the consumption by wildfire, the high $O_2$ consumption area is distributed in the low latitude tropics. These areas are rich in surface vegetation and have a high net primary production. The burning of vegetation will emit more carbon and thus consume more $O_2$. Savanna fires have the widest distribution and consume the most $O_2$, accounting for 60%~70% of the total; followed by tropical rain forests. Since the beginning of the 21st century, the total amount of $O_2$ consumption caused by the combustion process has been slowly decreasing.

When anthropogenic flux, including fossil fuel combustion, respiration of human and livestock, and wildfire are combined, several significant high $O_2$ consumption area like East Asia, India, eastern North America, Europe and central Africa are observed. Except for central Africa, these areas are densely populated with high levels of human activities and thus consume more $O_2$ than other areas of the world, while in central Africa, where the wildfire accounts for the vast majority of local $O_2$ consumption, also exhibits a high level of anthropogenic $O_2$ flux.

Then the global NEE data were used to establish the global terrestrial $O_2$ budget. It shows that due to the increasing fossil fuel combustion, overgrazing and population growth, current $O_2$ consumption over land is far greater than $O_2$ production from the terrestrial ecosystem, breaking the atmospheric $O_2$ balance and causing the decline of $O_2$ concentration in the atmosphere. During the period of 2000~2013, total anthropogenic $O_2$ flux over land has raised from 35.6 Gt/yr in 2000 to 46.0 Gt/yr in 2013, while the production from land (11.5Gt/yr averaged from 2000 to 2013) displays a faint increase during the same period. The regions where anthropogenic $O_2$ flux exceeds the biological $O_2$ flux and humans are consuming more $O_2$ than the ecosystem can provide are distributed in eastern Asia, Europe, North America, and northern South America, covering more than half of the land.

Finally, the land $O_2$ budgets in different regions are provided. In the tropics, the fossil fuel $O_2$ flux experienced a steadily growing trend while the wildfire flux indicates a downward trend, which may be related to the reduction in wildfire activities in recent decades. Temperate regions of the northern hemisphere consume the most $O_2$, accounting for than half of the global anthropogenic $O_2$ flux. In the boreal region of the northern hemisphere, wildfire activities are frequent and consume more $O_2$

than fossil fuel combustion. The Tibetan Plateau and boreal region of the northern hemisphere are other sources of atmospheric $O_2$ under the impact of human activities. Biological flux is playing a leading role in the local $O_2$ budget due to low levels of human activities.

**Author Contribution**

All authors contributed to shaping up the ideas and reviewing the paper. X.L and J.H contributed to the ideas and manuscript

writing. The calculation and analysis are carried out by Xiaoyue Liu.

**Competing interests**

The author declares that they have no conflict of interest.

**Acknowledgments**

This work was jointly supported by the National Science Foundation of China (41521004) and the China University

Research Talents Recruitment Program (the "111 project", No. B13045). The authors thank all people and institutions who provided the data used in this paper; the Scripps $O_2$ Program of the Scripps Institution of Oceanography for providing atmospheric $O_2$ data and; Carbon Dioxide Information Analysis Center (CDIAC) for providing the carbon emission data on both grid and country level; the NOAA ESRL, Boulder, Colorado, USA for providing the CarbonTracker CT2017 results from the website at http://carbontracker.noaa.gov; Global Fire Emission Database (GFED) for providing fire data. We thank

Julia Steinbach and Christoph Gerbig for providing the COFFEE dataset.

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




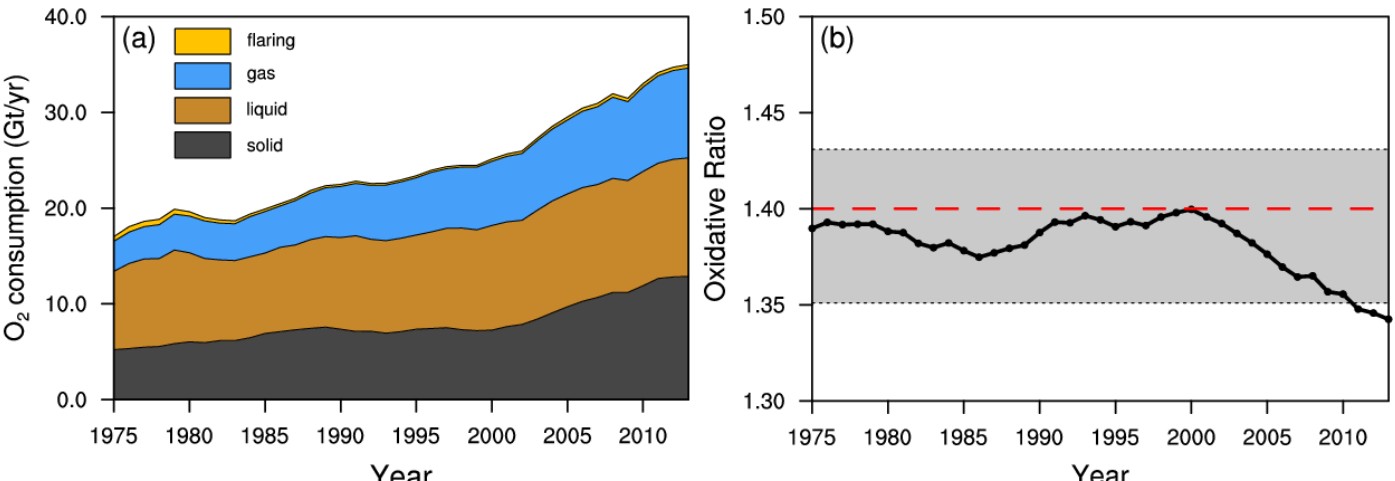

**Figure 2: Global total O₂ consumption by fossil fuel combustion and oxidative ratio from 1975 to 2013. Long-term trends of global total O₂ consumption by fossil fuel combustion (a) and global averaged oxidative ratio (b) from 1975 to 2013.**




Open Access    Earth System
Science
Data    Discussions

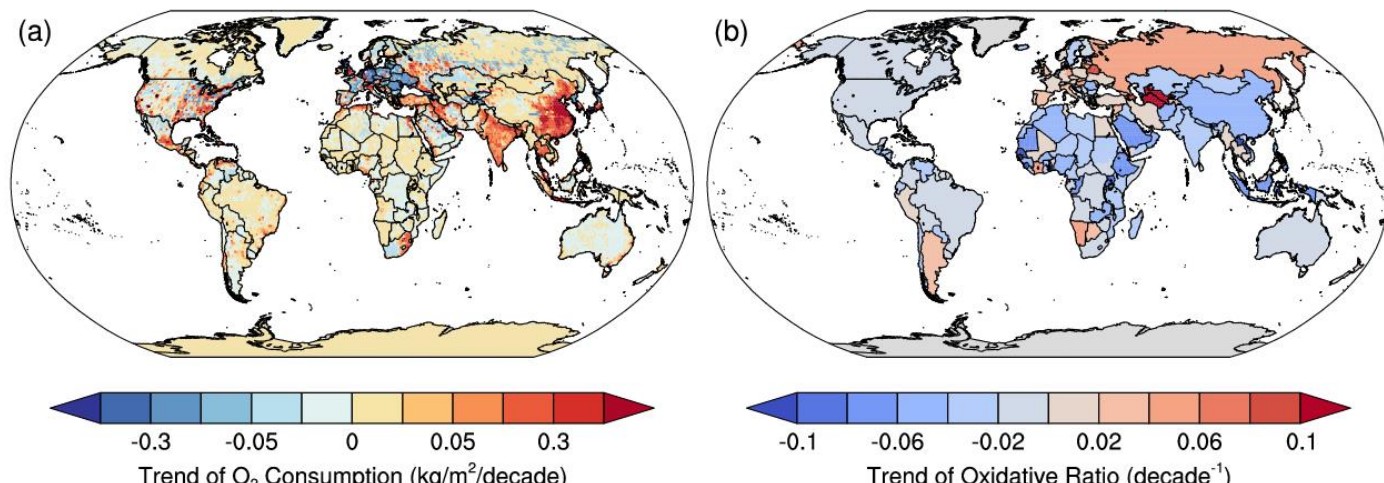

**Figure 3: Global patterns of the long-term trend of O₂ consumption by fossil fuel combustion and oxidative ratio from 1975 to 2013. Global pattern of long-term trends of global total O₂ consumption by fossil fuel combustion (a) and Global pattern of long-term trends of the global oxidative ratio (b) from 1975 to 2013.**





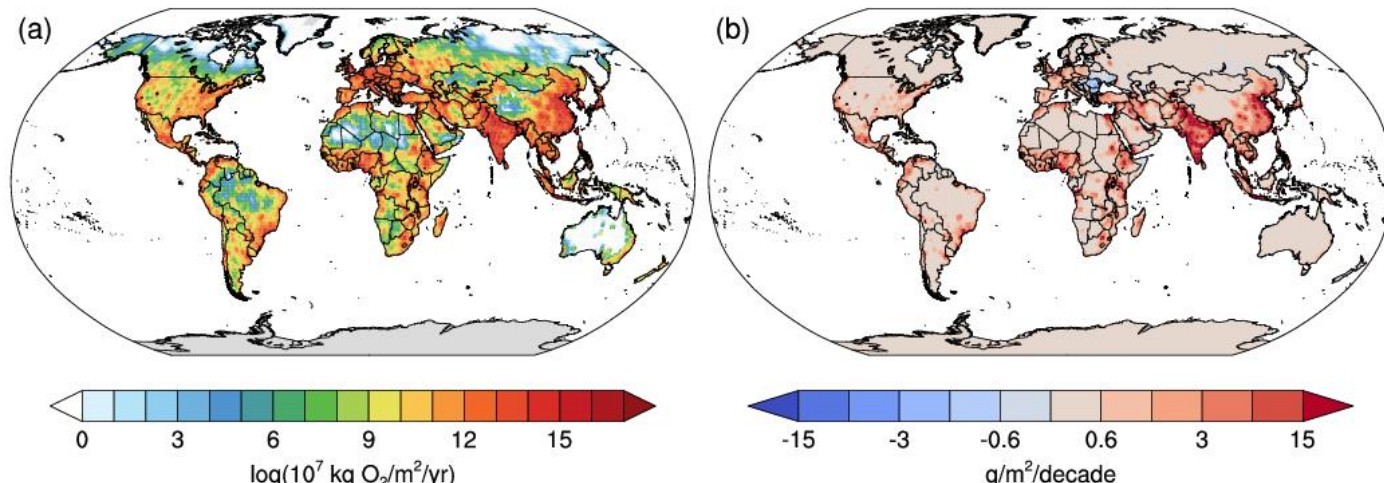

**Figure 4: Global pattern of O₂ consumption by human respiration and its long-term trends. O₂ consumption by human (a) for the year of 2013 and long-term trends (b) from 1975 to 2013.**



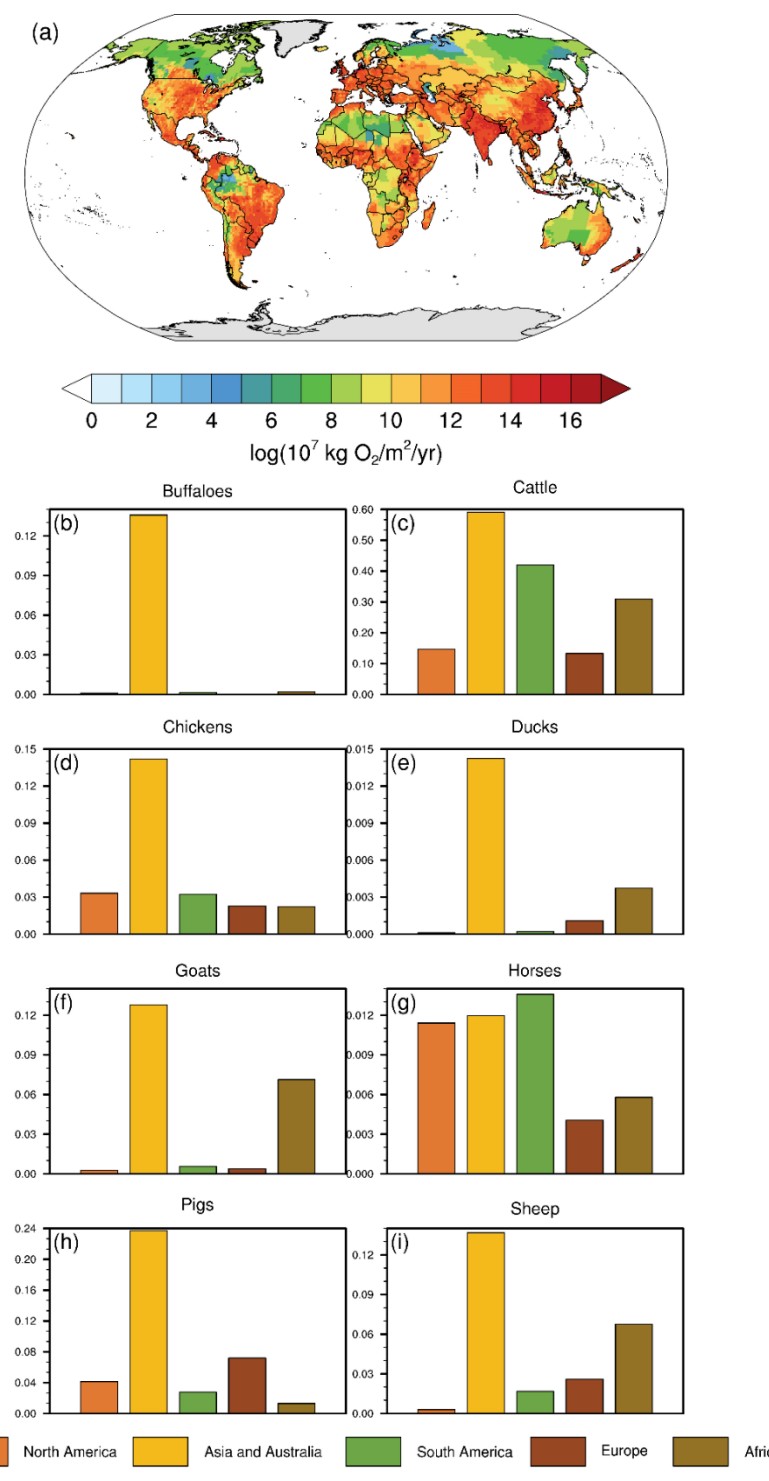

**Figure 5: Global pattern of O₂ consumption by livestock respiration. O₂ consumption by livestock (a) for the year of 2013 (b) Bar chart of O₂ consumed by 8 types of main livestock in each continent (Unit: Gt/yr).**



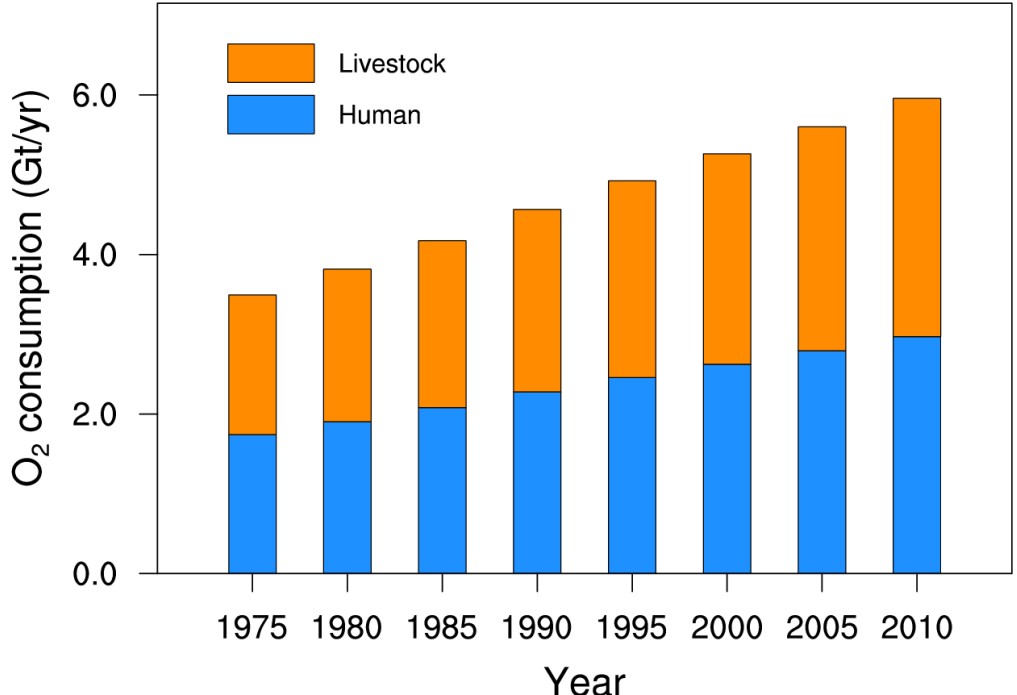

**Figure 6: Global total O₂ consumption by human and livestock respiration from 1975 to 2010 (Unit: Gt/a). The blue bar denotes the consumption by human and the orange bar denotes the consumption by livestock.**





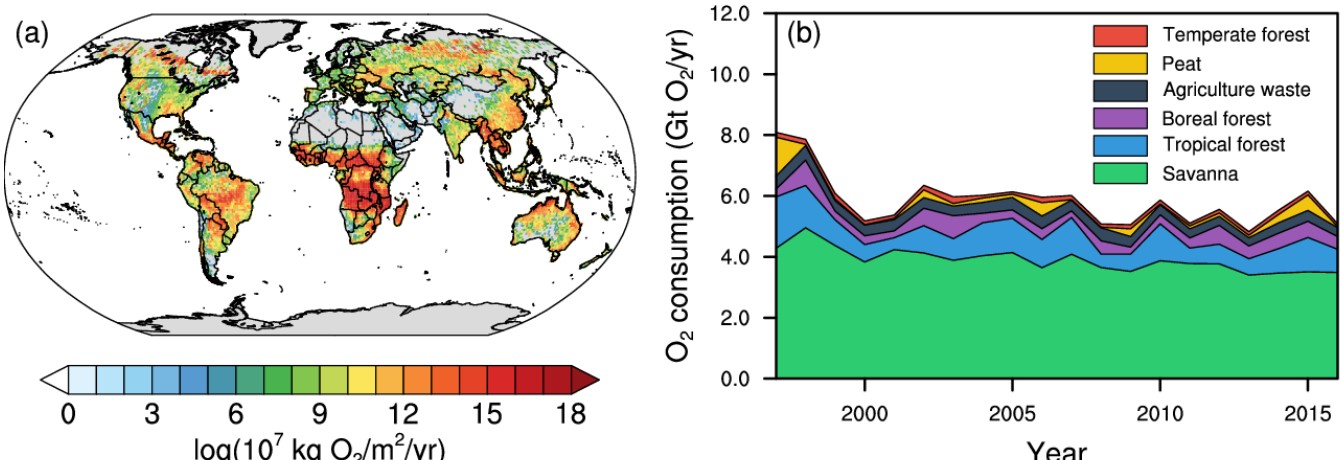

**Figure 7: Global O₂ consumption wildfire from 1997 to 2016. (a) O₂ consumption by human wildfire for the year 2013. (b) Long-term trends of global total O₂ consumption by different fire types from 1997 to 2016.**





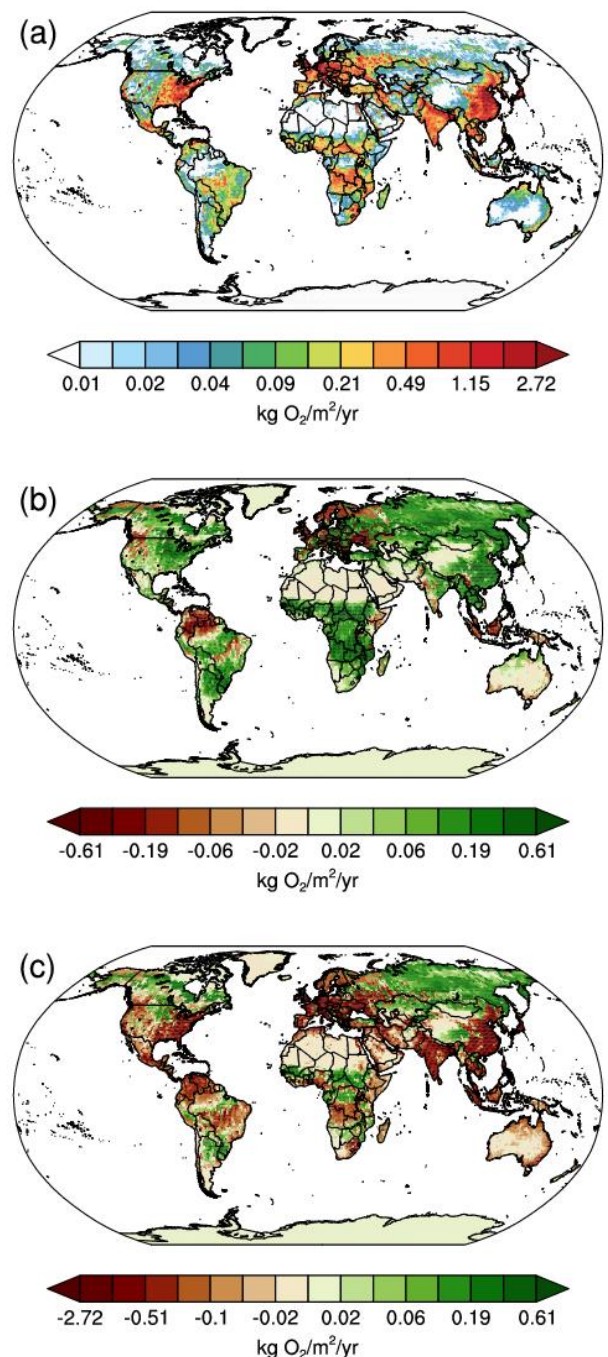

**Figure 8: Global land net O₂ flux from 2000 to 2013. (a) Anthropogenic O₂ flux including fossil fuel combustion, respiration of human and livestock, and wildfire averaged for the period of 2000-2013. Positive flux means the processes that remove the O₂ from the atmosphere. (b) Natural O₂ flux averaged for the period of 2000-2013. Positive flux means the processes that transport O₂ to the atmosphere and vice versa. (c) The difference between anthropogenic flux and natural flux averaged for the period of 2000-2013. The color bar is modified for better visualization.**



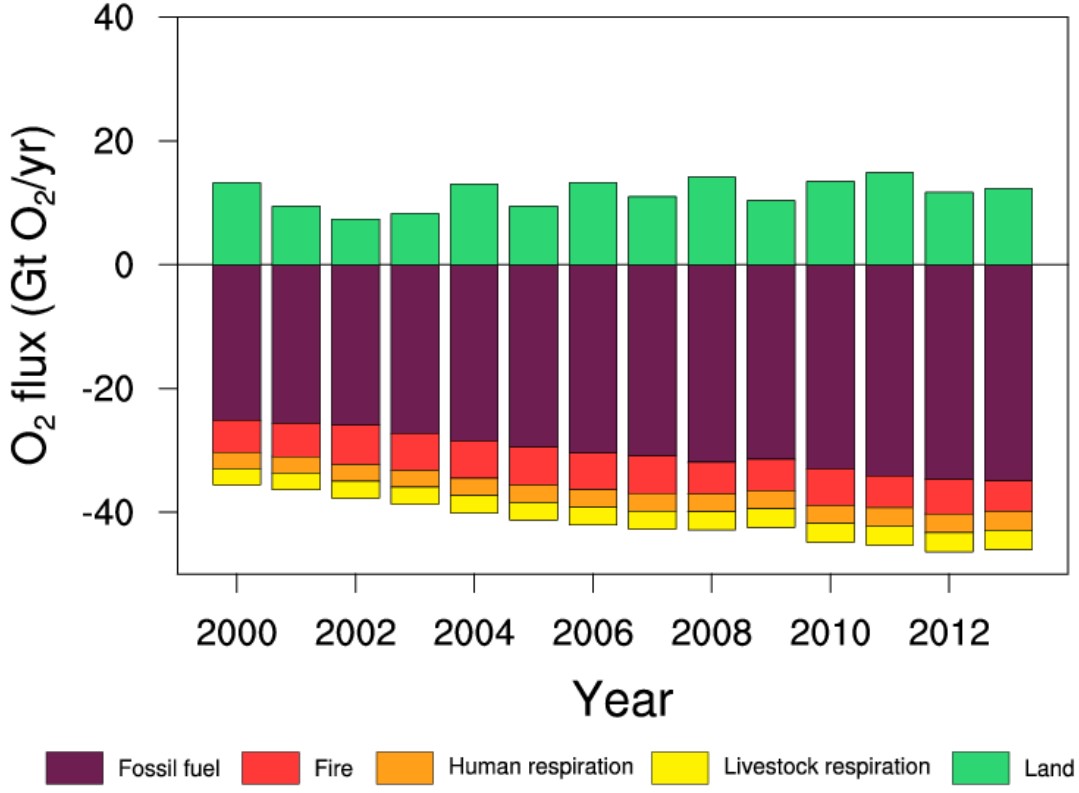

**Figure 9: Global total natural and anthropogenic O₂ flux from 2000 to 2013, with each component illustrated.**



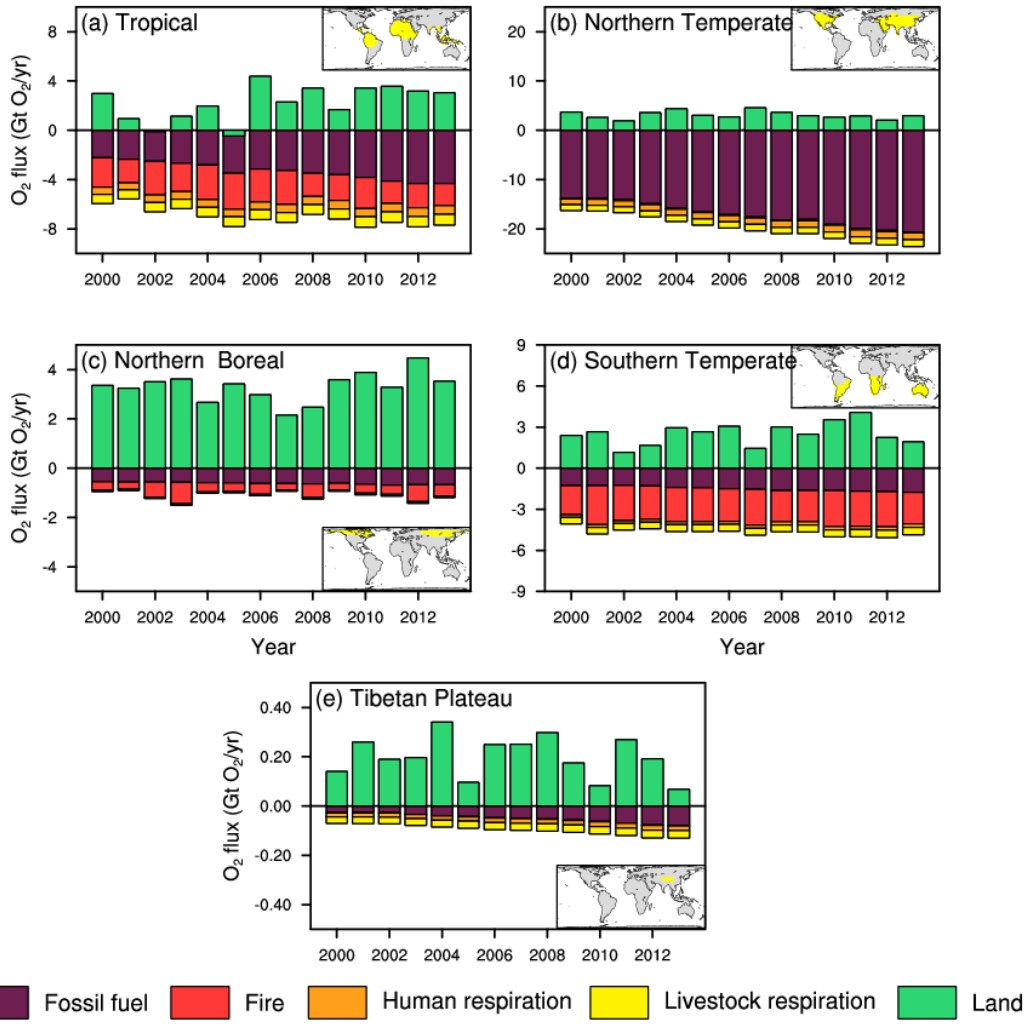

**Figure 10: Global land net O₂ flux in different regions from 2000 to 2013. (a) Tropical region. (b) Northern Temperate region. (c) Northern boreal region. (d) Southern temperate region. (e) Tibetan Plateau regions with an altitude greater than 3000m. The maps at the corner of each figure represent the calculated area.**



**Table 1. The oxidative ratio for each fuel type(Keeling, 1988)**

| Fuel type | $O_2$: $CO_2$ molar ratio |
| --- | --- |
| Solid fuel | 1.17 |
| Liquid fuel | 1.44 |
| Gas fuel | 1.95 |
| Flared gas | 1.98 |

