# Peer review of "Impact of anthropogenic activities on global land oxygen flux"

_Earth System Science Data, 2019_

## Referee Comment (RC1) · Anonymous Referee #1 · 3 Jun 2019

Review ESSD-2019-36, global land 02 flux

Possibly a useful topic for ESSD, to provide an alternative look at human emissions. But manuscript as submitted not at all up to standards expected by ESSD for useful global data sets. It seems the clear that the authors modified this manuscript from a prior research article. But the manuscript itself has not achieved successful transition from research article to data description.

O2 data in the manuscript lack any indication of uncertainty. All numbers and graphs presented as exact. Impossible. Read the ESSD guidance on presenting and discussing uncertainties. I hypothesize (and authors can prove me wrong) that uncertainties for human and livestock respiration (e.g. Figure 9) exceed the values as given.

Data set presented without validation. None! Again, please read ESSD guidelines on validation. Difficult here perhaps, but not impossible? By stoichiometry from the carbon budget (these authors attempt that but fail to apply such calculations as independent validation), but comparisons with other <population/energy/emissions> data bases could help. What about recent (e.g. lake, coastal) sediments with high deposition / time resolution? Oxygen isotopes could help? Authors must show that they have made best efforts on validation.

I read the global fire emission data differently than these authors, both for spatial and temporal patterns. These authors could prove the validity of their own calculations, but they give us no useful data or guidance.

A reader / user want confidence from these data, in order to use the data in their own work. The authors have given us no basis for such confidence.

Please can the authors read, carefully, these three ESSD publications:

Guidlines
https://doi.org/10.5194/essd-10-2275-2018

Global carbon budget (the authors cited it wrongly) for example of uncertainties, etc.)
https://doi.org/10.5194/essd-10-2141-2018

Global fire emissions database - the authors have not cited it correctly nor, I think, extracted valid conclusions from this data
https://doi.org/10.5194/essd-9-697-2017

The manuscript includes numerous technical and textual errors. I will read the manuscript again if the authors make the serious revisions suggested above. With their figures (e.g. 2a, 7b), these authors apparently aspire to emulate the global carbon budget but they have not demonstrated necessary levels of quality and reliability. If they hope to serve a global community with this data set, that community needs a much better data description.

---

## Referee Comment (RC2) · Anonymous Referee #2 · 6 Jun 2019

This paper takes maps carbon fluxes from fossil-fuel burning, fires, and net land exchanges and rescales them using conventional factors to produce O2 flux maps. Additionally, the paper presents maps of O2 loss from human and livestock respiration using assumptions about populations and metabolic rates.

I fail to see how these products are of value. There is no imminent threat of significant atmospheric O2 loss, such that tracking O2 for its own sake is an important environmental issue. The introduction to this paper has a sentence that gives a contrary impression, and is therefore quite misleading. The fluxes add little or nothing to our understanding of the global O2 budget. The maps of O2 loss from fossil-fuel burning are essentially those of Steinbach et al. (2011).

Another misleading element is the inclusion of human and livestock respiration in the

balances. These fluxes are not of primary importance for the global carbon balance, and similarly cannot be important for O2 balance. The fluxes are part of short-term loops which conserve CO2 and O2 overall. All food is derived from recent photosynthesis. I'm not aware of these fluxes being important in any context other than "horizontal displacement" as discussed, e.g. in Ciais et al (2008).

Ciais, P., A. V. Borges, G. Abril, M. Meybeck, G. Folberth, D. Hauglustaine and I. A. Janssens (2008). The lateral carbon pump, and the European carbon balance. The Continental-Scale Greenhouse Gas Balance of Europe, Springer: 341-360.

---

## Author Comment (AC1) · 12 Jun 2019

Thank you for your comments. The Oxygen cycle and Carbon cycle are actually two independent cycles. Although they are dependent on each other by some reaction processes such as respiration and photosynthesis (Keeling and Manning, 2014), they also act separately via other processes including photolysis of water, oxidation of minerals, etc., in which carbon doesn't involve (Berner, 2006; Petsch, 2013; Royer, 2013). Therefore, the dataset of the global Oxygen cycle should be necessary. As an integral part in the research of the Oxygen cycle, the oceanography community issued the Kiel Declaration (https://www.ocean-oxygen.org/declaration), addressing that the ongoing loss of oxygen is a rapidly increasing threat to marine ecosystems and improved understanding of its causes and consequences is immediately required. In addition, in

the field of geology, scientists are also making their effort to investigate the mechanism of The Great Oxidation Event (GOE) (Bekker et al., 2004; Canfield, 2014; Kump, 2008; Lyons et al., 2014) and rebuild the evolution of atmospheric oxygen during geologic period (Berner, 2009; Krause et al., 2018; Royer, 2013). The Oxygen cycle deserves the attention of the science community and the research on the global Oxygen cycle should be valued. The balance of oxygen sources and sinks involves an enormous array of diverse interactions between various "spheres" in the Earth System, including the biosphere (life), atmosphere (air), hydrosphere (water), lithosphere (rock) and anthroposphere (human). A complex web of chemical interactions of different timescales between the "spheres" have ultimately stabilized the atmospheric oxygen concentration during the past hundreds of millions of years. Oxygen is the only element that involves in all of the 5 "spheres" in the Earth System. In this paper, we aim to establish a global oxygen budget as comprehensive as possible. Therefore, the inclusion of human and livestock respiration is an essential part of the oxygen budget. Indeed, we have been aware of the fact that there is no imminent threat of significant atmospheric O2 loss. Even though the decline of O2 doesn't seem to bring about significant adverse effects on the survival of human for now, no one can assure that it will not pose a serious threat in the future. The imbalance between the CO2 and O2 budget has revealed the existence of the issue. Therefore, the research on the global O2 budget, along with the O2 flux data proposed in our paper, is of great value. Although this dataset is not perfect, we have made the first step. We hope that the data provided in this paper should excite the interest of users in a broad range of multi-disciplines (e.g. environmental science, bioscience, climate change, etc.).

Bekker, A., Holland, H. D., Wang, P. L., Rumble, D., Stein, H. J., Hannah, J. L., Coetzee, L. L. and Beukes, N. J.: Dating the rise of atmospheric oxygen, Nature, doi:10.1038/nature02260, 2004. Berner, R. A.: GEOCARBSULF: A combined model for Phanerozoic atmospheric O2 and CO2, Geochim. Cosmochim. Acta, 70(23 SPEC. ISS.), 5653–5664, doi:10.1016/j.gca.2005.11.032, 2006. Berner, R. A.: Phanerozoic atmospheric oxygen: new results using the geocarbsulf model,

Am. J. Sci., 309(7), 603–606, doi:10.2475/07.2009.03, 2009. Canfield, D. E.: Oxygen:A Four Billion Year History, Princeton University Press. [online] Available from: http://www.jstor.org/stable/j.ctt4cgd8h, 2014. Keeling, R. F. and Manning, A. C.: Studies of Recent Changes in Atmospheric O2 Content, 2nd ed., Elsevier Ltd., 2014. Krause, A. J., Mills, B. J. W., Zhang, S., Planavsky, N. J., Lenton, T. M. and Poulton, S. W.: Stepwise oxygenation of the Paleozoic atmosphere, Nat. Commun., 9(1), 1–10, doi:10.1038/s41467-018-06383-y, 2018. Kump, L. R.: The rise of atmospheric oxygen, Nature, 451(7176), 277–278, doi:10.1038/nature06587, 2008. Lyons, T. W., Reinhard, C. T. and Planavsky, N. J.: The rise of oxygen in Earth's early ocean and atmosphere, Nature, 506(7488), 307–315, doi:10.1038/nature13068, 2014. Petsch, S. T.: The Global Oxygen Cycle, 2nd ed., Elsevier Ltd., 2013. Royer, D. L.: Atmospheric CO2 and O2 During the Phanerozoic: Tools, Patterns, and Impacts, 2nd ed., Elsevier Ltd., 2013.

---

## Editor Comment (EC1) · David Carlson (Editor) · 18 Jun 2019

Global land O2 cycle and budget possibly an interesting research topic, but that debate should occur within the research literature. In view of two very negative assessments of this data product by ESSD reviewers, i suggest the authors withdraw the manuscript. Otherwise, in view of substantial deficiencies of the data presentation and description, almost certainly a rejection.